# Best-of-three-worlds Analysis for Dueling Bandits with Borda Winner

## Abstract

The dueling bandits (DB) problem addresses online learning from relative preferences, where the learner queries pairs of arms and receives binary win-loss feedback. Most existing work focuses on designing algorithms for specific stochastic or adversarial environments. Recently, a unified algorithm has been proposed that achieves convergence across all settings. However, this approach relies on the existence of a Condorcet winner, which is often not achievable, particularly when the preference matrix changes in the adversarial setting. Aiming for a more general Borda winner objective, there currently exists no unified framework that simultaneously achieves optimal regret across these environments. In this paper, we explore how the follow-the-regularized-leader (FTRL) algorithm can be employed to achieve this objective. We investigate a hybrid negative entropy regularizer and demonstrate that it enables us to achieve $\tilde{O}(K^{1/3}T^{2/3})$ regret in the adversarial setting, $O(K \log^2 T/\Delta_{\min}^2)$ regret in the stochastic setting, and $O(K \log^2 T/\Delta_{\min}^2 + (C^2 K \log^2 T/\Delta_{\min}^2)^{1/3})$ regret in the corrupted setting, where $K$ is the arm set size, $T$ is the horizon, $\Delta_{\min}$ is the minimum gap between the optimal and sub-optimal arms, and $C$ is the corruption level. These results align with the state-of-the-art in individual settings, while eliminating the need to assume a specific environment type. We also present experimental results demonstrating the advantages of our algorithm over baseline methods across different environments.

## 1 Introduction

In online sequential decision making, the multi-armed bandit framework (MAB) has played a crucial role in optimizing decisions under uncertainty (Lattimore & Szepesvári, 2020). Traditional MAB relies on absolute numerical rewards, which can often be noisy or challenging to obtain from users. To overcome this limitation, the dueling bandits (DB) problem provides a robust alternative using relative comparisons, where the learner queries pairs of actions (arms) and receives binary feedback on the preferred option (Bengs et al., 2021). This approach closely mirrors real-world scenarios where comparative judgments are more natural and reliable and has broad applications in areas such as search optimization, tournament ranking, retail management, and reinforcement learning from human feedback (RLHF) (Yue & Joachims, 2009; Dudik et al., 2011; Christiano et al., 2017).

DB algorithms aim to minimize regret over a given horizon, defined as the cumulative gap between the rewards of designated winners and the rewards obtained. The Condorcet and Borda winners are among the most widely studied winner objectives (Yue et al., 2012; Bengs et al., 2021). Existing research has explored various preference settings, including the stochastic case where relative preferences are fixed (Yue & Joachims, 2009), the adversarial case with arbitrarily changing preferences(Saha et al., 2021), and the corrupted case, which lies between them (Agarwal et al., 2021).

Despite the importance of these contributions, they provide convergence guarantees only under specific environments. Once the environment shifts—for instance, when an algorithm tailored for the stochastic setting is applied to an adversarial one—the performance can degrade to linear regret. Designing algorithms that achieve optimal performance across environments without relying on prior knowledge has therefore become a problem of broad interest in the field (Bubeck & Slivkins, 2012; Zimmert & Seldin, 2021; Kong et al., 2023; Tsuchiya et al., 2023; Ito & Takemura, 2023b;a; Ito et al., 2022).

Table 1: Regret bounds comparison under different environments and winners, where $K$ is the number of arms, $T$ is the time horizon, $C$ is the corruption level, $\Delta_{\min}$ is the minimum sub-optimality gap (to the optimal arm). $\tilde{O}$ hides polylogarithmic factors.

| | Environment | | |
| --- | --- | --- | --- |
| **Algorithm** | **Adversarial** | **Stochastic** | **Corrupted Stochastic** |
| **Condorcet** | | | |
| Saha & Gaillard (2022) *Versatile-DB* | $O(\sqrt{KT})$ | $O\left(\frac{K\log T}{\Delta_{\min}}\right)$ | $O\left(\frac{K\log T}{\Delta_{\min}} + \sqrt{K} + C\right)$ |
| **Borda** | | | |
| Saha et al. (2021) *Dueling-EXP3* | $\tilde{O}(K^{1/3}T^{2/3})$ | — | — |
| Saha et al. (2021) *BCB* | — | $O\left(\frac{K\log(KT)}{\Delta_{\min}^2}\right)$ | — |
| Ours | $\tilde{O}(K^{1/3}T^{2/3})$ | $O\left(\frac{K\log T\log(KT)}{\Delta_{\min}^2}\right)$ | $O\left(\frac{K\log T\log(KT)}{\Delta_{\min}^2} + \left(\frac{C^2 K\log T\log(KT)}{\Delta_{\min}^2}\right)^{1/3}\right)$ |

Saha & Gaillard (2022) study the best-of-three-world problem for dueling bandits under the Condorcet winner. They propose a DB–MAB reduction framework and demonstrate that existing analyses for MAB can be adapted to yield best-of-three-world guarantees for the DB setting. However, the Condorcet winner—defined as the arm that is preferred over every other arm with probability greater than 0.5 may not always exist, particularly in adversarial environments where the preference matrix evolves over time. By contrast, the Borda winner, defined as the arm that maximizes the average preference probability over all other arms, always exists regardless of the environment. Nevertheless, extending Saha & Gaillard (2022)'s DB–MAB reduction to the Borda winner is challenging because the regret definitions fundamentally differ: Condorcet regret decomposes into dominance gaps that enable a clean MAB mapping, whereas Borda regret aggregates average scores, rendering the Theorem 2 in Saha & Gaillard (2022)—which assumes pairwise independence for regret bounds—inapplicable from the outset. Therefore, establishing a best-of-three-world analysis for dueling bandits under the Borda winner remains an open problem.

In this paper, we address these gaps by developing an FTRL-based framework that directly optimizes the Borda score of the selected arms, rather than reducing the problem to a standard MAB formulation. Our approach leverages a hybrid negative entropy regularizer and demonstrates that the proposed algorithm simultaneously achieves performance guarantees across different environments. Specifically, we establish regret upper bounds of $\tilde{O}(K^{1/3}T^{2/3})$ in the adversarial setting, $O(K\log^2 T/\Delta_{\min}^2)$ in the stochastic setting, and $O(K\log^2 T/\Delta_{\min}^2 + (C^2 K\log^2 T/\Delta_{\min}^2)^{1/3})$ in the corrupted setting, where $K$ is the arm set size, $T$ is the horizon, $\Delta_{\min}$ is the minimum gap between the optimal and sub-optimal arms, and $C$ is the corruption level. Table 1 summarizes these results and compares our guarantees with those of existing works. To the best of our knowledge, this is the first algorithm to achieve a best-of-three-worlds guarantee for the dueling bandit problem under the general Borda winner objective. Moreover, our algorithm achieves the optimal $\tilde{O}(K^{1/3}T^{2/3})$ adversarial regret of Saha et al. (2021) and retains the key $\tilde{O}(K/\Delta_{\min}^2)$ scaling for stochastic environments. We also provide empirical validation on different environments where our algorithm demonstrates consistent advantages over baselines for a fixed environment type.

## 2 RELATED WORK

Research has mostly targeted specific settings—stochastic, adversarial, and corrupted stochastic—along with key winner types, such as the Condorcet winner (an arm that beats all others more than half the time) and the Borda winner (an arm with the highest average preference score). In stochastic settings with Condorcet winners, where preferences stay constant, algorithms like RUCB perform well under Condorcet winners by balancing exploration and exploitation (Zoghi et al., 2014); fur-

ther progress includes approaches that test for winners to identify them more efficiently in low-noise cases (Haddenhorst et al., 2021) and versatile methods that work across stochastic and adversarial worlds while keeping strong stochastic guarantees (Saha & Gaillard, 2022). For Borda winners in stochastic DB, techniques based on generalized linear models help estimate the full preference matrix to reduce regret (Wu et al., 2024), and adaptations for non-stationary environments use weighted Borda scores to handle slight changes over time (Suk & Agarwal, 2024b). Moving to adversarial settings, where preferences can shift unpredictably, MAB-style reductions allow algorithms to handle Condorcet winners robustly (Saha & Gaillard, 2022). For Borda winners here, the Dueling-Exp3 algorithm delivers strong results even without a Condorcet winner existing (Saha et al., 2021), and multi-dueling versions manage interactions between dependent arms (Gajane, 2024). In corrupted environments, which mix stable stochastic preferences with bounded adversarial noise, robust methods like Winner Isolation with Recourse protect against disruptions for Condorcet winners (Agarwal et al., 2021), and studies of attacks show how stochastic setups can be vulnerable (Jun et al., 2018); however, no tailored approaches yet exist for Borda winners in this mixed setting. Other winner concepts, like Copeland winners (which maximize direct wins against others) for handling cycles in preferences (Zoghi et al., 2015) and Von Neumann winners (mixed strategies that tie or beat all pure arms) for contextual scenarios (Di et al., 2025), broaden the framework further. Overall, while these works advance DB in isolated cases, a unified best-of-three-worlds solution for Borda winners— delivering top performance without knowing the environment in advance—is still missing, which inspires our FTRL-based method.

Best-of-both-worlds (BoBW) and best-of-three-worlds (BoTW) algorithms deliver near-optimal regret without prior environment knowledge, adapting across stochastic, adversarial, and corrupted settings. In multi-armed bandits, foundational BoBW methods introduce algorithms that perform well in both stochastic and adversarial regimes by integrating exploration mechanisms (Bubeck & Slivkins, 2012), while subsequent work achieves nearly optimal pseudo-regret bounds for these settings (Auer & Chiang, 2016; Zimmert & Seldin, 2021). Notably, (Zimmert & Seldin, 2021) introduces an optimal FTRL-based algorithm using Tsallis entropy regularization, providing tight pseudo-regret bounds in both stochastic and adversarial regimes. For linear bandits, BoBW designs attain near instance-optimality in stochastic cases and minimax-optimality in adversarial ones using optimistic online mirror descent with loss estimators (Lee et al., 2021), or exploration-by-optimization to balance exploration and optimization (Ito & Takemura, 2023a). In BoTW for linear bandits, which incorporates corrupted environments, follow-the-regularized-leader (FTRL) with negative entropy regularization and self-bounding analysis yields adaptive regret across all three worlds (Kong et al., 2023), and variance-adaptive algorithms tune bounds hierarchically to noise levels in stochastic, corrupted, and adversarial regimes (Ito & Takemura, 2023b). For linear contextual bandits, BoBW methods provide near-optimal regret in stochastic and adversarial settings via debiased estimators and FTRL with tailored perturbations (Kuroki et al., 2024). In dueling bandits, BoBW analyses via multi-armed bandit reductions offer guarantees under Condorcet winners for stochastic and adversarial preferences (Saha & Gaillard, 2022); yet, no BoTW frameworks exist, especially for Borda winners, creating a gap in unified adaptation that our FTRL approach fills.

## 3 PROBLEM SETTING

We study the problem *dueling bandits*, an online decision-making framework that involves a set of $K$ items, denoted by $[K] = \{1, 2, \ldots, K\}$, over a time horizon of $T$ rounds. At the beginning of the process, the environment determines a sequence of preference matrices $M_1, M_2, \ldots, M_T$ in advance, where each $M_t \in [0,1]^{K \times K}$ encodes the pairwise preference probabilities in round $t \in [T] := \{1, 2, \ldots, T\}$. Each matrix $M_t$ satisfies the following structural properties: $M_t(i,j) = 1 - M_t(j,i)$ for all $i, j \in [K]$, and $M_t(i,i) = \frac{1}{2}$ for all $i \in [K]$. Here, $M_t(i,j)$ represents the probability that item $i$ beats item $j$ in a pairwise comparison in round $t$. At each round $t \in [T]$, the learner selects two distinct items $x_t, y_t \in [K]$, and observes stochastic feedback $f_t \sim \text{Bernoulli}(M_t(x_t, y_t))$, where $f_t = 1$ indicates that item $x_t$ wins, and $f_t = 0$ indicates that item $y_t$ wins. To evaluate item quality, we introduce the *Borda score* of item $i \in [K]$ at round $t$, defined as $\sigma_t(i) = \frac{1}{K-1} \sum_{j=1, j \neq i}^{K} M_t(i,j)$, which measures the average probability that item $i$ wins against a randomly chosen distinct item. And the Borda winner $i^* \in [K]$ is the item with the highest cumulative Borda score over all rounds: $i^* = \arg\max_{i \in [K]} \sum_{t=1}^{T} \sigma_t(i)$.

The learner's performance is quantified by the *total regret*: $R_T = \sum_{t=1}^{T} \rho_t$, where $\rho_t = \sigma_t(i^*) - \frac{1}{2}(\sigma_t(x_t) + \sigma_t(y_t))$, which compares the Borda score of the Borda winner with the average score of the items chosen by the learner at each round.

For convenience, we also define the *shifted Borda score* for item $i \in [K]$ at round $t$ as $w_t(i) = \frac{1}{K} \sum_{j=1}^{K} M_t(i, j)$, which includes the self-comparison term $M_t(i, i) = \frac{1}{2}$. The corresponding *shifted regret* is $R_T^b = \sum_{t=1}^{T} \left[ w_t(i^*) - \frac{1}{2}(w_t(x_t) + w_t(y_t)) \right]$, where $i^*$ is the same Borda winner as defined above.

This is a shifted version of the original Borda score $\sigma_t(i) = \frac{1}{K-1} \sum_{j \neq i} M_t(i, j)$, where the summation now includes the term $M_t(i, i) = \frac{1}{2}$. The relationship between them is $w_t(i) = \frac{K-1}{K} \sigma_t(i) + \frac{1}{2K}$, which does not change the identity of the optimal item or the proportionality of the regret (with $R_T = \frac{K}{K-1} R_T^b$).

The shifted Borda score is defined and used primarily to simplify the construction of unbiased estimates for item scores in adversarial dueling bandits problem with Borda winner (Saha et al., 2021). In our algorithm, estimates are derived from binary preference feedback on pairs sampled i.i.d. with replacement from a distribution $d_t$. Including self-comparisons in $w_t(i)$ allows for symmetric and straightforward expectation calculations, avoiding the need to exclude self-pairs (which would complicate sampling without replacement and increase variance).

## 3.1 PREFERENCE REGIMES

We define the stochastic, adversarially corrupted stochastic, and adversarial environments for dueling bandits using the self-bounding constraint framework from Zimmert & Seldin (2021). These adapt standard multi-armed bandit models to pairwise comparisons, fitting dueling bandits. They unify regret analysis across adversarial levels, as in Zimmert & Seldin (2021). An environment follows a self-bounding constraint with $(\Delta, C, T)$ if, for any algorithm,

$$R_T^b \geq \mathbf{E}\left[ \sum_{t=1}^{T} \Delta(I_t) - C \right], \tag{1}$$

where $\Delta : [K] \to [0, 1]$. Here, $I_t$ is a representative arm sampled from the algorithm's distribution $\pi_t$, and since $x_t$ and $y_t$ are independently and identically distributed from $\pi_t$, the average score over the pair is equivalent to the performance of a single arm $I_t$.

**Stochastic Environments:** This is a special case with a $(\Delta, 0, T)$ self-bounding constraint (Zimmert & Seldin, 2021), where $\Delta(i) = w(i^*) - w(i)$ for a fixed distribution $\mathcal{D}$ over scores $w_t$.(In the stochastic dueling bandits setting, the Borda score of any arm/item is a fixed deterministic constant that remains identical across all rounds) Scores $w_t$ are drawn independently from $\mathcal{D}$ for each $t$, and the (pseudo-)regret satisfies the inequality with $C = 0$.

**Adversarially Corrupted Stochastic Environments with Corruption Level $C$:** This is a case with a $(\Delta, 2C, T)$ self-bounding constraint (Zimmert & Seldin, 2021), where $C \geq 0$ is the total corruption budget (adjusted for the factor of 2 from bounded regret differences). $\Delta$ is as in the stochastic case for some $\mathcal{D}$. Scores $w_t$ satisfy $\sum_{t=1}^{T} \max_i |w_t(i) - w_t'(i)| \leq C$ for $w_t' \sim \mathcal{D}$ independently per $t$. When $C = 0$, it reduces to stochastic.

**Adversarial Environments:** This covers all adversarial settings as a regime with a $(\Delta, 2T, T)$ self-bounding constraint (Zimmert & Seldin, 2021) for any $\Delta : [K] \to [0, 1]$. Scores $w_t$ are chosen arbitrarily by an adversary without assumptions. The constraint is vacuous with $C = 2T$ (from bounded losses in $[0, 1]$ and max deviation), including all adversarial cases.

These regimes span settings for regret analysis, from adversarial to stochastic.

## 4 ALGORITHM

In this section, we introduce the proposed follow-the-regularized-leader algorithm for dueling bandits in adversarial, stochastic, and corrupted environments, and present its pseudocode in 1.

In this approach, we define a probability distribution $\pi_t$ over $\Delta[K]$ (the $K$-simplex), as follows:

$$d_t \in \arg\max_{p \in \Delta[K]} \left\{ \sum_{s=1}^{t-1} \langle \hat{u}_s, p \rangle - \phi_t(p) \right\}, \pi_t = (1 - \delta_t)d_t + \delta_t U_K, \tag{2}$$

Where $U_K$ is an uniform distribution which satisfies $U_K(i) = 1/K$. And for initialization, set $d_1$ as the uniform distribution with each component equal to $\frac{1}{K}$.

---

**Algorithm 1** FTRL for Dueling Bandits

---

**Require:** Regularizers $\{\phi_t\}_t$, parameters $\{\delta_t\}_t$, arm count $K$

1: **for** each round $t = 1, 2, \ldots, T$ **do**
2:    Draw $x_t, y_t \overset{\text{i.i.d.}}{\sim} \pi_t$
3:    Observe $f_t(x_t, y_t) \sim \text{Ber}(M_t(x_t, y_t))$
4:    Set $\hat{u}_t(i) \leftarrow \frac{\mathbf{1}\{x_t=i\}}{K\pi_t(i)} \sum_{j \in [K]} \frac{\mathbf{1}\{y_t=j\}f_t(x_t,y_t)}{\pi_t(j)}$ for all $i \in [K]$
5:    $d_t \leftarrow \arg\max_{p \in \Delta(K)} \left\{ \sum_{s=1}^{t-1} \langle \hat{u}_s, p \rangle - \phi_t(p) \right\}$
6:    $\pi_t \leftarrow (1 - \delta_t)d_t + \delta_t u_K$
7: **end for**

---

The algorithm computes a distribution $\pi_t$ over the $K$-simplex $\Delta(K) = \{p : [K] \rightarrow [0, 1] \mid \sum_{i \in [K]} p(i) = 1\}$ in each round $t$, as defined in 2 from the original formulation. Firstly, it proceeds to independently sample $x_t$ and $y_t$ from $\pi_t$ 2. The preference feedback $f_t(x_t, y_t)$ follows a Bernoulli distribution with parameter $M_t(x_t, y_t)$ 3. Based on observations, the unbiased estimator $\hat{u}_t : [K] \rightarrow \mathbb{R}$ updates as shown in 4. Then it solves for $d_t$ by maximizing the sum of inner products with prior unbiased estimators minus a regularizer 5, where $\langle a, b \rangle = \sum_{i \in [K]} a(i)b(i)$, and $\phi_t : \Delta(K) \rightarrow \mathbb{R}$ is a convex Legendre function. Next, $\pi_t$ mixes $d_t$, using $\delta_t \in [0, 0.5]$ ( 6).

## 5 REGRET-BOUND ANALYSIS

Using the regularizer defined in 3, we can gain the regret bound in 1. We consider the regularizer functions defined as

$$\phi_t(p) = \alpha_t \sum_{i \in [K]} g(p(i)), \text{where} \quad g(x) = x \ln x + (1 - x) \ln(1 - x), \tag{3}$$

where the parameters $\delta_t$ and $\alpha_t$ are defined by $\alpha_1 = \max\{m_2, 8K\}$ and

$$\delta'_t = \frac{1}{4} \frac{m_1 v_t}{m_1 + \left(\sum_{s=1}^t v_s\right)^{1/3}}, \quad \alpha_{t+1} = \alpha_t + \frac{m_2 v_t}{\delta'_t \left(m_1 + \sum_{s=1}^{t-1} \frac{v_s e_{s+1}}{\delta'_s}\right)^{1/2}}, \quad \delta_t = \delta'_t + \sqrt[3]{\frac{K}{\alpha_t}}, \tag{4}$$

with $m_1, m_2 > 0$ as input parameters such that $m_1 \geq 2 \log K$ (used for computing $\delta'_t$ and $\alpha_t$ to ensure lower bounds) and $m_2 > 0$ (used to initialize $\alpha_1 = \max\{m_2, 8K\}$ and update $\alpha_{t+1}$),

Additionally, $\{e_t\}, \{v_t\}$ are defined by

$$e_t = -\sum_{i \in [K]} g(d_t(i)), \quad v_t = \sum_{i \in [K]} d_t(i)(1 - d_t(i)).$$

**Remark 1.** *The step sizes are designed to enable the algorithm to adapt automatically to the underlying environment. In the adversarial setting, to achieve the $T^{2/3}$ regret bound, we set $\delta_t \sim t^{-1/3}$, $\alpha_t \sim t^{2/3}$, and $\alpha_{t+1} - \alpha_t \sim t^{-1/3}$, as these rates ensure that the regret terms in Lemma 4 balance appropriately when $v_t$ and $e_t$ remain constant. In the stochastic setting, where $v_t$ and $e_t$ decrease from non-zero values to zero as the distribution concentrates on the optimal arm, the step sizes are chosen such that the regret can be controlled through the $\bar{S}^{2/3}$ term. By applying the inequality $x^{1/3}y^{2/3} \leq \frac{1}{3}x + \frac{2}{3}y$ in the self-bounding analysis (cf. equation equation 20), we derive the logarithmic regret bound of $O(\log T / \Delta_{min}^2)$.*

**Theorem 1.** *For any $T$, 1 with $\phi_t$, $\delta_t$, and $\alpha_t$ defined by 3 and 4 enjoys a regret bound of*

$$R_T \leq \bar{k} \cdot \max\left\{\bar{S}^{2/3}, m_1^2\right\}, \quad \text{where}$$

$$\bar{k} = O\left(m_1 + \frac{1}{\sqrt{m_1}}\left(\frac{K \log T}{m_2} + m_2\right)\sqrt{\log(KT)}\right). \tag{5}$$

*Consequently, if $T \geq K^3$, we have $R_T = O\left(\bar{k}T^{2/3}\right)$ in the adversarial regime and*

$$R_T = O\left(\frac{\bar{k}^3}{\Delta_{\min}^2} + \left(\frac{C^2\bar{k}^3}{\Delta_{\min}^2}\right)^{1/3}\right) \tag{6}$$

*in adversarial regimes with self-bounding constraints.*

This implies: $\tilde{O}(K^{1/3}T^{2/3})$ for adversarial environments, $O(\frac{K \log T \log(KT)}{\Delta_{\min}^2})$ for stochastic environments, and $O(\frac{K \log T \log(KT)}{\Delta_{\min}^2} + (\frac{C^2 K \log T \log(KT)}{\Delta_{\min}^2})^{1/3})$ for corrupted stochastic environments.

**Discussions.** The core challenge in extending best-of-both-worlds (BoBW) or best-of-three-worlds (BoTW) analyses to dueling bandits under the Borda winner benchmark lies in the fundamental mismatch between existing frameworks, such as the DB–MAB reduction in Saha & Gaillard (2022), and the inherent global nature of Borda scores. Saha & Gaillard (2022) provides a unified algorithm achieving optimal regrets across stochastic and adversarial environments under the Condorcet winner (CW) assumption, relying on a regret definition that decomposes into dominance gaps (e.g., $\mathbb{E}[R_T] = \frac{1}{2}\mathbb{E}[R_{-1,T} + R_{+1,T}]$. In their setting where $R_{-1,T}$ and $R_{+1,T}$ are the regrets achieved by the two multi-armed bandit algorithms corresponding to the two duelists, where $\Delta(i,j) = P(i,j) - 1/2$ to enable clean mapping to independent MAB instances. However, this fails for Borda winners because the regret definitions fundamentally differ: CW regret leverages pairwise independence via uniform dominance, whereas Borda regret aggregates average scores ($\sigma_t(i^*) - \frac{1}{2}(\sigma_t(x_t) + \sigma_t(y_t))$), rendering the Theorem 2 in Saha & Gaillard (2022)—which assumes such decomposition for regret bounds—inapplicable from the outset. This disparity arises from Borda's lack of uniform dominance, necessitating global preference matrix estimation that introduces coupled dependencies across all pairs. To circumvent this, we adopt an FTRL approach, directly optimizing over the simplex to embrace Borda's global averages.

Our FTRL algorithm uses a hybrid negative entropy regularizer $\phi_t(p) = \alpha_t \sum_i g(p(i))$, where $g(x) = x \ln x + (1-x)\ln(1-x)$, to address problems when using the standard Shannon entropy ($\phi_t(p) = \alpha_t \sum_i p(i) \ln p(i)$). In the stability bound 2, Shannon entropy produces a positive quadratic term in the Taylor expansion for small $s$ ($\sum_i p(i)s(i) + \frac{1}{2}\sum_i p(i)s(i)^2/\alpha_t + O(s^3/\alpha_t^2)$), which disrupts the recursive closure of the self-bounding inequality in 1 and fails to control the process quantity $v_t = \sum_i d_t(i)(1 - d_t(i))$. This stalls the analysis, as it cannot maintain the recursive structure needed for unified regret bounds. Our regularizer, however, yields a negative quadratic term ($\sum_i p(i)s(i) - \frac{1}{2}p(i)s(i)^2/\alpha_t + O(s^3/\alpha_t^2)$), which enables tight recursive control of $v_t$, ensuring the self-bounding inequality closes effectively across all regimes.

**Proof of the main theorem.** The proof relies on a series of lemmas and a key proposition, which we present as they are used to establish the result.

We begin by introducing the proposition that bounds a specific regret term $R_T^a$.

**Proposition 1.** *We begin by introducing the proposition that bounds a specific regret term $R_T^a$, which is an auxiliary regret term based on the shifted Borda score. Let us define parameters $\delta_t$ and $\alpha_t$ as in 1, then $R_T^a$ satisfying 10 is bounded as*

$$R_T^a = O\left(\mathbf{E}\left[m_1 V_T^{2/3} + \tilde{k}\sqrt{m_1^2 + (\log K + E_T)\left(m_1 + V_T^{1/3}\right)}\right]\right), \tag{7}$$

*where $E_T = \sum_{t=1}^{T} e_t$, $V_T = \sum_{t=1}^{T} v_t$, and $\tilde{k} = O\left(\frac{1}{\sqrt{m_1}}\left(\frac{K \log T}{m_2} + m_2\right)\right)$.*

This proposition is supported by the following lemmas which provide the intermediate bounds.

**Lemma 1.** *The Bregman divergence that we use is $D_{\phi_t}(p,q) = \phi_t(p) - \phi_t(q) - \langle \nabla \phi_t(q), p - q \rangle$. If $I_t$ is chosen following $\pi_t$ so that $\Pr[I_t = i \mid \pi_t] = \pi_t(i)$, the regret is bounded by*

$$R_T^b \leq \mathbf{E}\left[\sum_{t=1}^{T} \left(\delta_t - \langle \hat{u}_t, d_t - d_{t+1} \rangle - D_{\phi_t}(d_{t+1}, d_t) + \phi_t(d_{t+1}) - \phi_{t+1}(d_{t+1})\right)\right] \tag{8}$$
$$+ \phi_{T+1}(e_{i^*}) - \phi_1(d_1),$$

*where $e_{i^*}(i) = 1$ if $i = i^*$ and $e_{i^*}(i) = 0$ for $i \in [K] \setminus \{i^*\}$.*

**Lemma 2.** *When the traditional Shannon entropy is used as the regularizer, $\phi_t$ is defined as*

$$\phi_t(p) = -\alpha_t f(p), \quad \text{where} \quad f(p) = \sum_{i \in [K]} p(i) \ln \frac{1}{p(i)}.$$

*It holds for any $s : [K] \to \mathbb{R}$ and $p, q \in \mathcal{P}(K)$ that*

$$\langle s, q - p \rangle - D_{\phi_t}(q, p) \leq \alpha_t \sum_{i \in [K]} p(i) \zeta \left(\frac{-s(i)}{\alpha_t}\right),$$

*where $\zeta(x) = \exp(-x) + x - 1$.*

**Lemma 3.** *If $\phi_t$ is given by 3, it holds for any $s : [K] \to \mathbb{R}$ and $p, q \in \mathcal{P}(K)$ that*

$$\langle s, q - p \rangle - D_{\phi_t}(q, p) \leq \alpha_t \sum_{i \in [K]} \min \left\{ p(i) \zeta \left(\frac{-s(i)}{\alpha_t}\right), (1 - p(i)) \zeta \left(\frac{s(i)}{\alpha_t}\right) \right\}, \tag{9}$$

*where $\zeta(x) = \exp(-x) + x - 1$.*

**Lemma 4.** *Suppose $\phi_t$ is defined as in 3 and $\delta_t \geq \sqrt[3]{\frac{K}{\alpha_t}}$. Then, the regret satisfies $R_T \leq R_T^a + e_1 \alpha_1$, where*

$$R_T^a = O\left(\mathbf{E}\left[\sum_{t=1}^{T} \left(\delta_t + \frac{|K|v_t}{\delta_t \alpha_t} + (\alpha_{t+1} - \alpha_t) e_{t+1}\right)\right]\right), \tag{10}$$

*and the sequences $\{e_t\}$ and $\{v_t\}$ are given by*

$$e_t = -\sum_{i \in [K]} g\left(d_t(i)\right), \quad v_t = \sum_{i \in [K]} d_t(i) \left(1 - d_t(i)\right). \tag{11}$$

We use the above proposition along with the following lemma, which bounds the sums $E_T$ and $V_T$.

**Lemma 5.** *Consider the following definitions:*

$$E_T = \sum_{t=1}^{T} e_t, \quad V_T = \sum_{t=1}^{T} v_t, \quad \tilde{k} = O\left(\frac{1}{\sqrt{m_1}} \left(\frac{K \log T}{m_2} + m_2\right)\right), \tag{12}$$

*with input parameters $m_1, m_2 > 0$ satisfying $m_1 \geq 2 \ln K$. Then, for any $i^* \in [K]$, the sums $E_T$ and $V_T$ are bounded by*

$$E_T \leq 2S(i^*) \ln \frac{eKT}{S(i^*)}, \quad V_T \leq 2S(i^*), \tag{13}$$

*where $S(i^*)$ is as given in 17.*

From Proposition 1 (the bound in 7) and Lemma 5 (the bounds in 13), if $\bar{S} \geq m_1^3$, we have

$$R_T^a = O\left(\mathbf{E}\left[m_1 S(i^*)^{2/3} + \tilde{k}\sqrt{S(i^*) \log(KT) S(i^*)^{1/3}}\right]\right)$$
$$\leq O\left(\left(m_1 + \tilde{k}\sqrt{\log(KT)}\right) \bar{S}^{2/3}\right), \tag{14}$$

where the inequality follows from Jensen's inequality. Hence, there exists $\bar{k}$ such that,

$$R_T^a \leq \bar{k} \cdot \bar{S}^{2/3}$$
$$\text{where} \quad \bar{k} = O\left(m_1 + \tilde{k}\sqrt{\log(KT)}\right) \tag{15}$$

As a consequence, we obtain 5. Since $\bar{S} \leq T$, in adversarial regimes, it follows from 5 that

$$R_T^b = O\left(\bar{k} \cdot \max\left\{T^{2/3}, m_1^2\right\}\right) = O\left(\bar{k} \cdot T^{2/3}\right). \tag{16}$$

Let us next show 6. This relies on the following lemma, which provides a lower bound on the regret using self-bounding parameters.

**Lemma 6.** *We introduce the following parameters $S(i^*)$ and $\bar{S}$, which will be used when applying the self-bounding technique:*

$$S(i^*) = \sum_{t=1}^{T}(1 - d_t(i^*)), \quad \bar{S}(i^*) = \mathbf{E}[S(i^*)], \quad \bar{S} = \min_{i^* \in [K]} \bar{S}(i^*), \tag{17}$$

*We note that these values are clearly bounded as $0 \leq \bar{S} \leq \bar{S}(i^*) \leq T$ for any $i^* \in [K]$. In an adversarial regime with a self-bounding constraint, the regret can be bounded from below:*

$$R_T^b \geq \frac{\Delta_{\min}}{2}\bar{S} - C. \tag{18}$$

From 15 and Lemma 6, for any $\theta \in (0, 1]$, we have

$$R_T^b = (1 + \theta)R_T^b - \theta R_T^b = O\left((1 + \theta)\bar{k} \cdot \bar{S}^{2/3} - \theta\Delta_{\min}\bar{S} + \theta C\right). \tag{19}$$

We have

$$(1 + \theta)\bar{k} \cdot \bar{S}^{2/3} - \theta\Delta_{\min}\bar{S} = \left(\frac{(1 + \theta)^3\bar{k}^3}{\theta^2\Delta_{\min}^2}\right)^{1/3}\left(\theta\Delta_{\min}\bar{S}\right)^{2/3} - \theta\Delta_{\min}\bar{S}$$

$$= O\left(\frac{(1 + \theta)^3\bar{k}^3}{\theta^2\Delta_{\min}^2}\right) = O\left(\left(1 + \frac{1}{\theta^2}\right)\frac{\bar{k}^3}{\Delta_{\min}^2}\right), \tag{20}$$

where the second equality follows from $x^{1/3}y^{2/3} \leq \frac{1}{3}x + \frac{2}{3}y$ for any $x, y \geq 0$. Combining these inequalities, we obtain

$$R_T^b = O\left(\left(1 + \frac{1}{\theta^2}\right)\frac{\bar{k}^3}{\Delta_{\min}^2} + \theta C\right). \tag{21}$$

By choosing $\theta$ that minimizes the right-hand side, we obtain 6.

Setting $m_1 = \Theta\left((K \log T \cdot \log(KT))^{1/3}\right)$ and $m_2 = \Theta\left(\sqrt{K \log T}\right)$, we obtain

$$\bar{k} = O\left((K \log T \cdot \log(KT))^{1/3}\right). \tag{22}$$

Then we get that an algorithm achieves $R_T^b = \tilde{O}\left(K^{1/3}T^{2/3}\right)$ for adversarial environments, $R_T^b = O\left(\frac{K \log T \log(KT)}{\Delta_{\min}^2}\right)$ for stochastic environments, and $R_T^b = O\left(\frac{K \log T \log(KT)}{\Delta_{\min}^2} + \left(\frac{C^2 K \log T \log(KT)}{\Delta_{\min}^2}\right)^{1/3}\right)$ for adversarially-corrupted stochastic environments. Finally, since $R_T = \frac{K}{K-1}R_T^b$, this does not change the order of magnitude we have obtained. Therefore, we can subsequently arrive at the result in 1.

## 6 EXPERIMENTS

To represent an environment, we use a preference matrix, where the first row corresponds to the arm with the highest total value, the second row corresponds to the second-best arm, and so on (Figure 1 (a)). For the adversarial setting, we construct a reversed preference matrix (Figure 1 (b)), where the first row corresponds to the worst arm and the last row to the best.

The environment alternates between the original and reversed matrices: the algorithm learns on the original matrix for 100 rounds, then on the reversed one for 150 rounds. The extra 50 rounds help offset any residual influence from learning in the original environment. And due to the rearrangement inequality, this setting of the environment can cause the most severe regret when changing the

environment. In the corrupted setting, we modify the preference matrix by swapping the first and second rows (Figure 1 (c)) every 500 rounds.

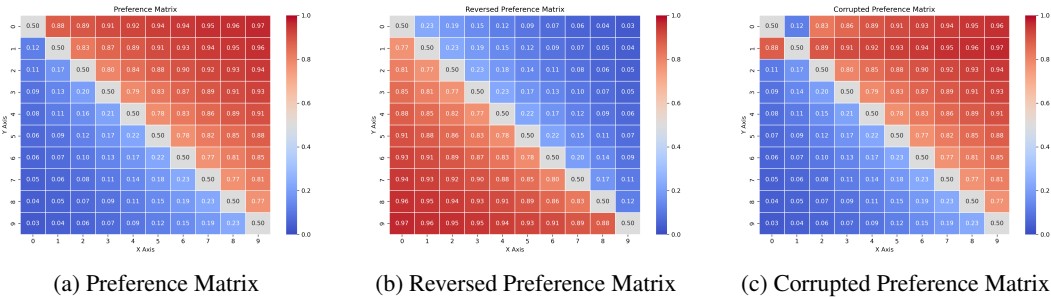

(a) Preference Matrix  (b) Reversed Preference Matrix  (c) Corrupted Preference Matrix

Figure 1: Experimental Setting of Three Preference Matrices.

For comparison, we evaluate two established algorithms for dueling bandits with Borda winner: Borda-Confidence-Bound (BCB) for stochastic environments, and Dueling-EXP3 (D-EXP3) for adversarial environments (Saha et al., 2021). In each experiment, the reported results are averaged over five independent runs. As shown in Figure 2, our algorithm achieves better performance than D-EXP3 in the stochastic setting, outperforms BCB in the adversarial setting, and surpasses both algorithms in the corrupted setting.

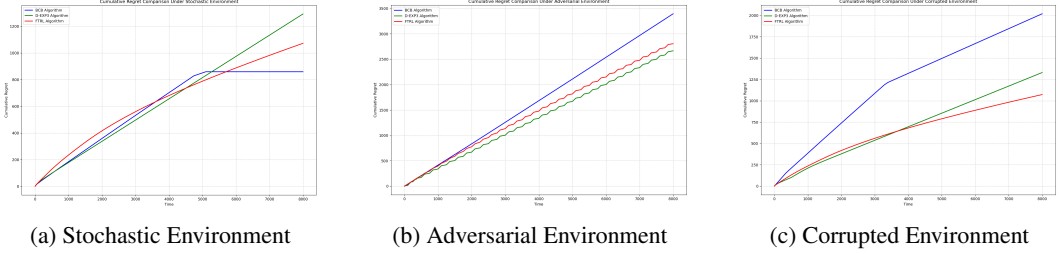

(a) Stochastic Environment  (b) Adversarial Environment  (c) Corrupted Environment

Figure 2: Experimental Results in Three Environments.

## 7 CONCLUSION AND FUTURE SCOPES

We address the dueling bandits problem under the Borda winner benchmark, where the goal is to minimize regret from relative preferences across stochastic, corrupted stochastic, and adversarial environments without prior knowledge of the setting. We overcome the core challenge of extending existing frameworks, such as the DB–MAB reduction tailored for Condorcet winners, which fails due to Borda's global averaging nature requiring full preference matrix estimation. Our key contributions include: (1) the first unified best-of-three-worlds (BoTW) framework for Borda winners, achieving nearly optimal regrets of $\tilde{O}(K^{1/3}T^{2/3})$ in adversarial, $O(K \log^2 T/\Delta_{\min}^2)$ in stochastic, and $O(K \log^2 T/\Delta_{\min}^2 + (C^2 K \log^2 T/\Delta_{\min}^2)^{1/3})$ in corrupted settings; (2) an FTRL algorithm with a hybrid negative entropy regularizer and time-varying rates for adaptive self-bounding; (3) empirical validation demonstrating superior robustness over baselines.

Our BoTW framework for Borda winners opens potential extensions to contextual dueling bandits, where preferences depend on side information (Dudík et al., 2015). By adapting our FTRL with hybrid regularization, we could explore unified BoTW regret bounds for stochastic, corrupted, and adversarial settings. In addition, our framework can be extended to human feedback reinforcement learning (RLHF), where dueling bandits model preference-based alignment (Christiano et al., 2017), potentially achieving robust BoTW guarantees under noisy or adversarial feedback by using our adaptive regularization approach.

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

## A SUPPLEMENTARY PROOF OF LEMMAS AND PROPOSITION IN THE MAIN TEXT

### A.1 LEMMA 1

If $I_t$ is chosen following $\pi_t$ so that $\Pr[I_t = i \mid \pi_t] = \pi_t(i)$, the regret is bounded by

$$
R_T \leq \mathbf{E}\left[\sum_{t=1}^{T}\left(\delta_t - \langle \hat{u}_t, d_t - d_{t+1}\rangle - D_{\phi_t}(d_{t+1}, d_t) + \phi_t(d_{t+1}) - \phi_{t+1}(d_{t+1})\right)\right] \\
+ \phi_{T+1}(e_{i^*}) - \phi_1(d_1),
$$

(23)

where $e_{i^*}(i) = 1$ if $i = i^*$ and $e_{i^*}(i) = 0$ for $i \in [K] \setminus \{i^*\}$.

*Proof.* From the definition of the algorithm, we have

$$
R_T(i^*) = \mathbf{E}\left[\sum_{t=1}^{T} w_t(i^*) - \sum_{t=1}^{T} w_t(I_t)\right] = \mathbf{E}\left[\sum_{t=1}^{T}\langle -w_t, \pi_t - e_{i^*}\rangle\right]
$$

$$
= \mathbf{E}\left[\sum_{t=1}^{T}\langle -w_t, d_t - e_{i^*}\rangle + \sum_{t=1}^{T}\delta_t\langle -w_t, u_K - d_t\rangle\right]
$$

$$
\leq \mathbf{E}\left[\sum_{t=1}^{T}\langle -w_t, d_t - e_{i^*}\rangle + \sum_{t=1}^{T}\delta_t\right]
$$

$$
= \mathbf{E}\left[\sum_{t=1}^{T}\langle -\hat{u}_t, d_t - e_{i^*}\rangle + \sum_{t=1}^{T}\delta_t\right],
$$

where the second equality follows from $I_t \sim \pi_t$, the inequality follows from $\langle -w_t, u_K - d_t \rangle \leq \langle -w_t, u_K \rangle \leq 1$, and the last equality follows from the fact that $\hat{u}_t$ is an unbiased estimator for $w_t$. Further, from Exercise 28.12 of the book by Lattimore & Szepesvári (2020), we have

$$-\sum_{t=1}^{T} \langle \hat{u}_t, d_t - e_{i^*} \rangle \leq \sum_{t=1}^{T} \left( -\langle \hat{u}_t, d_t - d_{t+1} \rangle - D_{\phi_t}(d_{t+1}, d_t) + \phi_t(d_{t+1}) - \phi_{t+1}(d_{t+1}) \right)$$
$$+ \phi_{T+1}(e_{i^*}) - \phi_1(d_1).$$

Combining this, we obtain the regret bound in 8. $\qquad\square$

## A.2  LEMMA 2

*Proof.* Consider the partial derivative of the left-hand side expression with respect to each $q(i)$:

$$\frac{\partial}{\partial q(i)} \left( \langle s, q - p \rangle - D_{\phi_t}(q, p) \right) = s(i) - \alpha_t(\ln q(i) - \ln p(i)).$$

Since the expression is concave in $q$, the maximum over $q \in \mathbb{R}_{>0}^K$ occurs where this derivative vanishes, namely at $q^*(i) = p(i) \exp\left( \frac{s(i)}{\alpha_t} \right)$. Therefore,

$$\langle s, q - p \rangle - D_{\phi_t}(q, p) \leq \langle s, q^* - p \rangle - D_{\phi_t}(q^*, p)$$
$$= \sum_{i \in [K]} \left[ s(i)(q^*(i) - p(i)) - \alpha_t \left( q^*(i) \ln q^*(i) - p(i) \ln p(i) \right) \right.$$
$$\left. - (q^*(i) - p(i))(\ln p(i) + 1) \right)]$$
$$= \sum_{i \in [K]} \left( -s(i)p(i) + s(i)q^*(i) - \alpha_t q^*(i) \ln q^*(i) + \alpha_t p(i) \ln p(i) \right.$$
$$\left. + \alpha_t(q^*(i) - p(i))(\ln p(i) + 1) \right)$$
$$= \sum_{i \in [K]} \left( -s(i)p(i) + \alpha_t(q^*(i) - p(i)) \right)$$
$$= \alpha_t \sum_{i \in [K]} p(i) \left( \exp\left( \frac{s(i)}{\alpha_t} \right) - 1 - \frac{s(i)}{\alpha_t} \right)$$
$$= \alpha_t \sum_{i \in [K]} p(i) \zeta \left( \frac{-s(i)}{\alpha_t} \right).$$

The initial equality stems from the Bregman divergence formula. The subsequent simplification uses $\ln q^*(i) = \ln p(i) + \frac{s(i)}{\alpha_t}$, and the later step substitutes $q^*(i) = p(i) \exp\left( \frac{s(i)}{\alpha_t} \right)$. This establishes the result for Lemma 2. $\qquad\square$

## A.3  LEMMA 3

*Proof.* Let us introduce a non-negative function $d(y, x)$ for $x, y \in (0, 1)$, given by

$$d(y, x) = y \ln \frac{y}{x} + x - y = y \ln y - x \ln x - (y - x)(\ln x + 1).$$

This $d$ represents the Bregman divergence on the interval $(0, 1)$ corresponding to the potential $\phi^{(1)}(x) = x \ln x$. When $\phi_t$ follows 3, its associated Bregman divergence $D_{\phi_t}(q, p)$ can be written as

$$D_{\phi_t}(q, p) = \alpha_t \sum_{i \in [K]} \left[ d(q(i), p(i)) + d(1 - q(i), 1 - p(i)) \right].$$

Consequently, we derive

$$
\begin{aligned}
\langle s, q - p \rangle - D_{\phi_t}(q, p) &= \sum_{i \in [K]} \left[ s(i)(q(i) - p(i)) - \alpha_t \left( d(q(i), p(i)) + d(1 - q(i), 1 - p(i)) \right) \right] \\
&\leq \sum_{i \in [K]} \min \left\{ s(i)(q(i) - p(i)) - \alpha_t d(q(i), p(i)), \right. \\
&\qquad\qquad \left. -s(i)(p(i) - q(i)) - \alpha_t d(1 - q(i), 1 - p(i)) \right\}.
\end{aligned}
$$

Drawing from the reasoning in the proof of lemma2 , it follows that

$$
s(i)(q(i) - p(i)) - \alpha_t d(q(i), p(i)) \leq \alpha_t p(i) \zeta \left( \frac{-s(i)}{\alpha_t} \right).
$$

Analogously, we can establish

$$
\begin{aligned}
&s(i)(q(i) - p(i)) - \alpha_t d(1 - q(i), 1 - p(i)) \\
&= s(i)((1 - p(i)) - (1 - q(i))) - \alpha_t \Big[ (1 - q(i)) \ln(1 - q(i)) - (1 - p(i)) \ln(1 - p(i)) \\
&\qquad - ((1 - q(i)) - (1 - p(i)))(\ln(1 - p(i)) + 1) \Big] \\
&\leq \alpha_t (1 - p(i)) \zeta \left( \frac{s(i)}{\alpha_t} \right).
\end{aligned}
$$

By integrating these, we arrive at the inequality stated in 9. This concludes the demonstration of Lemma 3. $\qquad\square$

### A.4   LEMMA 4

*Proof.* To establish this result, we start by applying Lemma 3 to analyze the term $\langle -\hat{u}_t, d_t - d_{t+1} \rangle - D_{\phi_t}(d_{t+1}, d_t)$. For every $t \in [T]$ and $i \in [K]$, we notice that $\hat{u}_t(i) \leq K/\delta_t^2$. This inequality arises from the expression for $\hat{u}_t(i)$ in  1, combined with the lower bound $\pi_t(i) \geq \delta_t/K$ for all $i \in [K]$. Since $\delta_t \geq (K/\alpha_t)^{1/3}$, it follows that $\alpha_t \delta_t^2 \geq K \delta_t^{-1}$, implying $\hat{u}_t(i) \leq \alpha_t \delta_t^{-1}$.

Using the inequality $\zeta(x) \leq x^2/2$ for $|x| \leq 1$, and noting that $|s(i)|/\alpha_t \leq \delta_t^{-1}$ for $s = \hat{u}_t$, we derive

$$
\begin{aligned}
\langle -\hat{u}_t, d_t - d_{t+1} \rangle - D_{\phi_t}(d_{t+1}, d_t) &\leq \alpha_t \sum_{i \in [K]} \min \left\{ d_t(i) \zeta \left( \frac{-\hat{u}_t(i)}{\alpha_t} \right), (1 - d_t(i)) \zeta \left( \frac{\hat{u}_t(i)}{\alpha_t} \right) \right\} \\
&\leq \frac{K v_t}{\delta_t \alpha_t}.
\end{aligned}
$$

where the last step uses the definition of $v_t$ in 11 and the bound $\zeta \left( \frac{\pm \hat{u}_t(i)}{\alpha_t} \right) \leq \frac{1}{2} \left( \frac{\hat{u}_t(i)}{\alpha_t} \right)^2 \leq \frac{1}{2\delta_t^2}$.

Next, we bound $\phi_t(d_{t+1}) - \phi_{t+1}(d_{t+1})$. Since $\phi_t(p) = \alpha_t \sum_{i \in [K]} g(p(i))$ and $g(x) < 0$ for $x \in (0, 1)$, we have

$$
\begin{aligned}
\phi_t(d_{t+1}) - \phi_{t+1}(d_{t+1}) &= (\alpha_t - \alpha_{t+1}) \sum_{i \in [K]} g(d_{t+1}(i)) = -(\alpha_{t+1} - \alpha_t) \sum_{i \in [K]} g(d_{t+1}(i)) \\
&= (\alpha_{t+1} - \alpha_t) e_{t+1},
\end{aligned}
$$

where the last equality follows from $e_{t+1} = -\sum_{i \in [K]} g(d_{t+1}(i))$.

Combining this with the bound from 8 in Lemma 1 and the inequality above, we obtain

$$
R_T \leq \mathbf{E} \left[ \sum_{t=1}^{T} \left( \delta_t + \frac{K v_t}{\delta_t \alpha_t} + (\alpha_{t+1} - \alpha_t) e_{t+1} \right) \right] + \phi_{T+1}(e_{i^*}) - \phi_1(d_1).
$$

Note that $\phi_{T+1}(e_{i^*}) = \alpha_{T+1} \sum_{i \in [K]} g(e_{i^*}(i)) = 0$, since $g(1) = 0$ and $g(0) = 0$. Additionally, $-\phi_1(d_1) = -\alpha_1 \sum_{i \in [K]} g(d_1(i)) = \alpha_1 e_1$. Therefore,

$$
R_T \leq R_T^a + e_1 \alpha_1,
$$

where $R_T^a$ is defined as in 10. This completes the proof. $\qquad\square$

## A.5 LEMMA 5

*Proof.* Since $d_t(i) \geq \delta_t/K \geq 1/(2K)$ for all $i \in [K]$ and $t \in [T]$, we have $v_t = \sum_{i \in [K]} d_t(i)(1 - d_t(i)) \leq 1 - d_t(i^*) - \sum_{i \neq i^*} d_t(i)^2 \leq 1 - d_t(i^*) - (K-1)\left(\frac{\delta_t}{K}\right)^2 \leq 1 - d_t(i^*) - \frac{1}{4K}$. Therefore,

$$1 - d_t(i^*) \leq v_t + \frac{1}{4K} \leq 2v_t,$$

which implies $S(i^*) \leq 2V_T$. Next, we bound $E_T$ using the concavity of $g$. From Jensen's inequality and the definition of $e_t = -\sum_{i \in [K]} g(d_t(i))$, we have

$$e_t \geq -Kg\left(\frac{1}{K}\right) = \ln\frac{eK}{1} + (K-1)\ln(K-1) - K\ln K \geq \ln(eK),$$

where the second inequality follows from $\ln(K-1) \geq \ln K - 1$. For the upper bound, note that $g(x) = x\ln x + (1-x)\ln(1-x) \leq x\ln x + (1-x)\ln(1-x) + x(1-x)$ and thus $-g(x) \leq -x\ln x - (1-x)\ln(1-x)$. Therefore,

$$\begin{aligned}
e_t &\leq \sum_{i \in [K]} \left[-d_t(i)\ln d_t(i) - (1-d_t(i))\ln(1-d_t(i))\right] \\
&= -d_t(i^*)\ln d_t(i^*) - (1-d_t(i^*))\ln(1-d_t(i^*)) \\
&\quad + \sum_{i \neq i^*} \left[-d_t(i)\ln d_t(i) - (1-d_t(i))\ln(1-d_t(i))\right] \\
&\leq -d_t(i^*)\ln d_t(i^*) - (1-d_t(i^*))\ln(1-d_t(i^*)) + \sum_{i \neq i^*} \ln\frac{1}{d_t(i)} \\
&\leq -d_t(i^*)\ln d_t(i^*) - (1-d_t(i^*))\ln(1-d_t(i^*)) + (K-1)\ln\left(\frac{1-d_t(i^*)}{K-1}\right) \\
&\leq 2(1-d_t(i^*))\ln\frac{e(K-1)T}{1-d_t(i^*)},
\end{aligned}$$

where the third inequality uses the concavity of $\ln$ and Jensen's inequality, and the last inequality follows from $-x\ln x \leq x$ for $x \in [0,1]$ and $d_t(i^*) \leq 1 - 1/T$. Summing over $t$ and using the bound above, we obtain the bounds in 13. $\qquad\square$

## A.6 LEMMA 6

*Proof.* From the definition of the regret and the self-bounding constraint, we have

$$R_T \geq \mathbf{E}\left[\sum_{t=1}^{T} \Delta(I_t) - C\right] \geq \Delta_{\min}\mathbf{E}\left[\sum_{t=1}^{T}(1 - d_t(i^*))\right] - C = \Delta_{\min}\bar{S}(i^*) - C,$$

where the second inequality follows from $\Delta(i) \geq \Delta_{\min}(1 - \mathbf{1}\{i = i^*\})$ and the fact that $\pi_t(i) = (1-\delta_t)d_t(i) + \delta_t/K \geq (1-\delta_t)d_t(i) \geq \frac{1}{2}d_t(i)$ due to the assumption of $\delta_t \leq \frac{1}{2}$ and the definitions of $S(i^*)$ and $\bar{S}$ in 17. $\qquad\square$

## A.7 PROPOSITION 1

*Proof.* Observe that $v_t \leq 1$ and $v_t \leq e_t \leq 2\log K$. Introduce the auxiliary variables $l_t = \frac{v_t e_{t+1}}{\delta_t'}$ and $L_t = \sum_{s=1}^{t} l_s$. By the definition of $\delta_t'$, it follows that

$$l_t = \frac{e_{t+1}v_t}{\delta_t'} = 4\frac{e_{t+1}}{m_1}\left(m_1 + V_t^{1/3}\right) \geq e_{t+1} \geq v_{t+1},$$

since $v_t \leq e_t$ implies the second inequality. Additionally,

$$l_t = 4\frac{e_{t+1}}{m_1}\left(m_1 + V_t^{1/3}\right) \leq 4\left(m_1 + V_t^{1/3}\right) \leq 4m_1 + 4\left(v_1 + \sum_{s=1}^{t-1} l_s\right)^{1/3}$$

$$\leq 8\left(m_1 + L_{t-1}\right),$$

where the initial inequality uses $e_{t+1} \leq 2 \log K$ combined with $m_1 \geq 2 \log K$, and the final step relies on $m_1 \geq 2$ and $v_1 \leq 1$. Consequently,

$$\sum_{t=1}^{T} (\alpha_{t+1} - \alpha_t) e_{t+1} = m_2 \sum_{t=1}^{T} \frac{l_t}{\sqrt{m_1 + L_{t-1}}} = 4m_2 \sum_{t=1}^{T} \frac{L_t - L_{t-1}}{3\sqrt{m_1 + L_{t-1}} + \sqrt{m_1 + L_{t-1}}}$$

$$\leq 4m_2 \sum_{t=1}^{T} \frac{L_t - L_{t-1}}{\sqrt{m_1 + L_t} + \sqrt{m_1 + L_{t-1}}}$$

$$= 4m_2 \sum_{t=1}^{T} \left( \sqrt{m_1 + L_t} - \sqrt{m_1 + L_{t-1}} \right) \leq 4m_2 \sqrt{L_T},$$

with the equality arising from the definitions of $\alpha_t$ and $l_t$, and the inequality stemming from the bound above.

Next, define $n_t = \frac{v_t}{\delta_t'}$ and $N_t = \sum_{s=1}^{t} n_s$. From the expression for $\delta_t'$,

$$n_t = \frac{v_t}{\delta_t'} = 4 \left( 1 + \frac{1}{m_1} V_t^{1/3} \right) \geq 4.$$

Moreover,

$$n_1 \leq 8, \quad n_{t+1} = 4 \left( 1 + \frac{1}{m_1} V_{t+1}^{1/3} \right) \leq 4 \left( 1 + \frac{1}{m_1} (V_t + 1)^{1/3} \right) \leq 2n_t,$$

$$n_t \leq 4 \left( 1 + t^{1/3} \right).$$

The parameter $\alpha_t$ admits the lower bound

$$\alpha_t = m_2 + m_2 \sum_{s=1}^{t-1} \frac{n_s}{\sqrt{m_1 + L_{s-1}}} \geq \frac{m_2}{\sqrt{m_1 + L_t}} \left( 1 + \sum_{s=1}^{t-1} n_s \right)$$

$$= \frac{m_2}{\sqrt{m_1 + L_t}} (1 + N_{t-1}) \geq \frac{m_2 t}{\sqrt{m_1 + L_t}},$$

utilizing the definition in the last step. Therefore,

$$\sum_{t=1}^{T} \frac{v_t}{\delta_t \alpha_t} \leq \sum_{t=1}^{T} \frac{v_t}{\delta_t' \alpha_t} \leq \sum_{t=1}^{T} \frac{\sqrt{m_1 + L_t}}{m_2} \frac{n_t}{1 + N_{t-1}} \leq \frac{\sqrt{m_1 + L_T}}{m_2} \sum_{t=1}^{T} \frac{n_t}{1 + N_{t-1}}$$

$$\leq O \left( \frac{\sqrt{m_1 + L_T}}{m_2} \log T \right),$$

where the concluding inequality derives from

$$\ln (1 + N_t) - \ln (1 + N_{t-1}) = \ln \frac{1 + N_t}{1 + N_{t-1}} = \ln \left( 1 + \frac{n_t}{1 + N_{t-1}} \right) \geq \frac{1}{5} \cdot \frac{n_t}{1 + N_{t-1}},$$

valid since $\ln(1 + x) \geq \frac{1}{5}x$ for $x \in [0, 8]$, and the bounds ensure $\frac{n_t}{1+N_{t-1}} \leq 8$ for every $t$.

Furthermore,

$$\sum_{t=1}^{T} \frac{1}{\alpha_t} \leq \sum_{t=1}^{T} \frac{\sqrt{m_1 + L_t}}{m_2 t} \leq \frac{\sqrt{m_1 + L_T}}{m_2} \sum_{t=1}^{T} \frac{1}{t} = O \left( \frac{\sqrt{m_1 + L_T}}{m_2} \log T \right).$$

Additionally,

$$\sum_{t=1}^{T} \delta_t' \leq \sum_{t=1}^{T} \frac{v_t}{1 + V_t^{1/3}} \leq \frac{3m_1}{2} \sum_{t=1}^{T} \left( V_t^{2/3} - V_{t-1}^{2/3} \right) \leq \frac{3m_1}{2} V_T^{2/3},$$

leveraging the relation $y^{2/3} - x^{2/3} \geq \frac{2}{3}(y-x)y^{-1/3}$ for $y \geq x > 0$. Integrating these yields

$$
\sum_{t=1}^{T} \left( \delta_t + \frac{K v_t}{\delta_t \alpha_t} + (\alpha_{t+1} - \alpha_t) e_{t+1} \right)
$$

$$
= \sum_{t=1}^{T} \left( \delta_t' + \sqrt[3]{\frac{K}{\alpha_t}} + \frac{K v_t}{\delta_t \alpha_t} + (\alpha_{t+1} - \alpha_t) e_{t+1} \right)
$$

$$
= O \left( m_1 V_T^{2/3} + \left( \frac{K \log T}{m_2} + m_2 \right) \sqrt{m_1 + L_T} \right)
$$

$$
= O \left( m_1 V_T^{2/3} + \left( \frac{K \log T}{m_2} + m_2 \right) \sqrt{m_1 + \sum_{t=1}^{T} \frac{e_{t+1}}{m_1} \left( m_1 + V_t^{1/3} \right)} \right)
$$

$$
= O \left( m_1 V_T^{2/3} + \frac{1}{\sqrt{m_1}} \left( \frac{K \log T}{m_2} + m_2 \right) \sqrt{m_1^2 + (\log K + E_T) \left( m_1 + V_T^{1/3} \right)} \right),
$$

where the third line applies the bound, and the final step uses $e_{T+1} \leq 2 \log K$. $\qquad\square$

## B    CONNECTIONS TO PARTIAL MONITORING FRAMEWORKS

In this section, we discuss the relationship between dueling bandits with Borda winner and partial monitoring games, highlighting how our approach connects to and differs from existing PM methodologies.

### B.1    CONCEPTUAL RELATIONSHIP BETWEEN DUELING BANDITS WITH BORDA WINNER AND PARTIAL MONITORING GAMES

While previous studies have explored connections between dueling bandits and partial monitoring (Kirschner et al., 2020; Suk & Agarwal, 2024a), none provide a complete treatment of the Borda winner setting. Kirschner et al. (2020) analyze the Condorcet winner case, while Suk & Agarwal (2024a) address a generalized Borda formulation but not specifically the standard Borda dueling bandits problem. Their Remark 3 suggests that such generalized Borda problems may be reduced to partial monitoring with non-global observability, leading to a worst-case regret bound of $\Omega(T)$, which is not informative for our setting. Neither analysis applies directly to the DB problem with Borda winner.

We address this gap by proposing a reduction that maps the Borda dueling bandits problem to a finite PM game and by characterizing the resulting PM structure. Our reduction scheme is constructed as follows. The action set consists of all arm pairs $a = \{i, j\}$ $(i < j)$ with $k = \binom{K}{2}$ total actions. The outcome set contains all possible tournament preference matrices $x \in \{0, 1\}^{\binom{K}{2}}$ with $d = 2^{\binom{K}{2}}$ outcomes. The feedback $\Phi_{a,x}$ reveals the winner of duel $\{i, j\}$ through $\{+1, -1\}$ signals. We define the Borda score as $b_x(i) = \frac{1}{K-1} \sum_{\ell \neq i} \mathbf{1}[i \succ_x \ell]$ and identify the optimal Borda arm $i^*(x) = \arg\max_i b_x(i)$. The loss function implements standard Borda regret: $L_{a,x} = b_x(i^*(x)) - \frac{1}{2}(b_x(i) + b_x(j))$.

Because the Borda score aggregates comparisons against all other arms, we prove that the resulting PM game is globally observable with parameter $k_\Pi = K - 1$, the weight function $w_e$ for any two neighboring Pareto optimal actions $a = \{i^*, p\}$ and $b = \{i^*, q\}$ as $w_e(c, \sigma) = -\frac{1}{2} \left( w^{(p)}(c, \sigma) - w^{(q)}(c, \sigma) \right)$, where $w^{(i)}(c, \sigma) = \frac{1}{K-1}$ if $c = \{i, \ell\}$ for some $\ell$ and $\sigma$ indicates $i$ wins, and 0 otherwise. This construction satisfies the global observability condition in Definition 1 of (Tsuchiya et al., 2023), since $\sum_{c=1}^{k} w_e(c, \Phi_{c,x}) = L_{a,x} - L_{b,x}$ for all $x \in [d]$, as required by equation (1) in their paper. And $c_G = \max\{1, k\|G\|_\infty\} \leq \binom{K}{2} \cdot 1/(2(K-1)) \approx K/4$.

Applying the global-observability guarantees of Tsuchiya et al. (2023) to our reduced game yields regret bounds of $O(K^{4/3} T^{2/3} \log T)$ for the adversarial setting and $O(K^2 \log^2 T / \Delta_{\min}^2)$ for the

stochastic setting. Both bounds incur an additional multiplicative factor of $K$ compared with our tailored results for DB with a Borda winner. Thus, while partial monitoring offers a useful conceptual perspective, applying its general guarantees to the Borda dueling bandits problem results in strictly suboptimal results.

### B.2 ALGORITHMIC AND ANALYTICAL DISTINCTIONS

Our approach also differs fundamentally from general partial monitoring methodologies. Although both employ FTRL-style algorithms, the choice of regularizer and the resulting analysis are substantially different. In globally observable PM games, the feedback matrix $\Phi$ provides strong stability guarantees: loss differences can be estimated with low variance, quantified by the game-dependent constant $c_G$. This structure allows Tsuchiya et al. (2023) to use the standard Shannon entropy regularizer to control the key intermediate term $\langle \hat{y}_t, q_t - q_{t+1} \rangle - D_t(q_{t+1}, q_t)$ as demonstrated in the proof of their Lemma 11.

In contrast, the Borda dueling bandits problem does not possess such global stability properties. The Borda score depends on comparisons with all other arms, and there is no analogue of the PM constant $c_G$ that would enable the same Shannon-entropy–based argument. As a result, the analytical steps used in general PM frameworks cannot be directly replicated in our setting. To overcome this problem, we employ the hybrid entropy regularizer to derive the intermediate upper bound shown in Lemma 3. This lemma is essential for both our adversarial and stochastic analyses and represents a key technical innovation beyond standard PM approaches.

## C LLM USAGE

In the draft of this paper, we utilized Grok 4. Specifically, Grok 4 was employed for language polishing to improve the clarity, grammar, and flow of the text; generating and formatting tables based on provided data and descriptions; and suggesting adjustments to the paper's layout and structure for better readability and organization. These uses were limited to editorial and presentational enhancements and did not involve generating original research ideas, technical content, proofs, or experimental designs. The authors take full responsibility for all content in the paper, ensuring its originality and scientific integrity.

