# OpenReview forum: "Best-of-three-worlds Analysis for Dueling Bandits with Borda Winner"
_ICLR.cc/2026/Conference — ICLR 2026 Conference Withdrawn Submission_

### Official Review · Reviewer_zTzf · 2025-10-31

**Soundness:** 3
**Presentation:** 3
**Contribution:** 3
**Rating:** 6
**Confidence:** 3

**Summary:**

This work establishes the first best-of-three-worlds algorithm for dueling bandits under the Borda winner objective, developing a single FTRL-based algorithm which achieves order-optimal regret in both stochastic and adversarial settings, as well as in settings with a corruption budget. Previously, such a result was shown for dueling bandits with Condorcet winner and regret notion, but the regret scales differently in such a setting and so the Borda result requires new innovations, namely a hybrid negative entropy regularizer.

**Strengths:**

* The paper shows the first best-of-three-worlds regret bound for Borda dueling bandits.
* The writing is generally clear and presentation is easy to follow.
* There are also experiments to validate theoretical results, i.e. superiority of their FTRL algorithm over other approaches for the various settings.

**Weaknesses:**

I think there could be more discussion of related works to better highlight the technical novelty of this work. In particular, there are other works [1]-[3] on getting best-of-both or best-of-three-worlds results in settings where the regret scales like $T^{2/3}$ that should be definitely be mentioned. My understanding is that the hybrid regularizer used in this work is required to adapt the self-bounding analysis for this type of regret scaling rather than the $\sqrt{T}$ rate seen in Condorcet dueling bandits. My main concern is whether this Borda dueling bandit problem can in fact be instantiated as a partial monitoring problem and inherit the results of [2]. For instance, it is known that Condorcet dueling bandits can be viewed as a partial monitoring problem in some instances (see mentions of dueling bandits in "Information Directed Sampling for Linear Partial Monitoring" Kirschner et al., 2020) and also in some generalized versions of the Borda problem (see Remark 3 in the cited paper Suk and Agarwal., 2024), or whether the adaptations required in the FTRL self-bounding analysis are already implicit in the works [1]-[3].

[1] Nearly Optimal Best-of-Both-Worlds Algorithms for Online Learning with Feedback Graphs, Ito et al., NeurIPS 2022.

[2] Best-of-Both-Worlds Algorithms for Partial Monitoring. Tsuchiya et al., ALT 2023.

[3] Simultaneously Learning Stochastic and Adversarial Bandits with General Graph Feedback. Kong et al., ICML 2022.

**Questions:**

Please see my main question above in "Weaknesses".

---

> ### Author Response · Authors · 2025-11-20
>
> We thank the reviewer zTzf for the valuable comments and suggestions. Please find our response below.
>
> - Comparison with Tsuchiya et al., ALT 2023
>
> We thank the reviewer for directing us to this literature. Below, we address the reviewer’s question by discussing (i) how our results relate to those of Tsuchiya et al (2023), and (ii) the key technical distinctions between the two approaches.
>
>
> 1. Conceptual relationship between dueling bandits (DB) with Borda winner and partial monitoring (PM) games.
>
> Although previous studies (Kirschner et al., 2020; Suk \& Agarwal, 2024) have mentioned connections between DB and PM, none provide a complete treatment of the Borda winner setting. Kirschner et al (2020) analyze the Condorcet winner case, while Suk \& Agarwal (2024) address a generalized Borda formulation but not specifically the standard Borda dueling bandits problem. Their Remark~3 suggests that such generalized Borda problems may be reduced to partial monitoring with non-global observability, leading to a worst-case regret bound of $\Omega(T)$, which is not informative for our setting. We fill this gap by proposing a reduction that maps the Borda dueling bandits problem to a finite PM game and by characterizing the resulting PM structure.
>
> Our reduction scheme is constructed as follows.
> The action set consists of all arm pairs $a = \{i,j\}$ ($i < j$) with $k = \binom{K}{2}$ total actions.
> The outcome set contains all possible tournament preference matrices $x \in \{0,1\}^{\binom{K}{2}}$ with $d = 2^{\binom{K}{2}}$ outcomes.
> The feedback $\Phi_{a,x}$ reveals the winner of duel $\{i,j\}$ through $\{+1, -1\}$ signals.
>
> We define the Borda score as $b_x(i) = \frac{1}{K-1} \sum_{\ell \neq i} \mathbf{1}[i \succ_x \ell]$ and identify the optimal Borda arm $i^*(x) = \arg\max_i b_x(i)$.
>
> The loss function implements standard Borda regret: $L_{a,x} = b_x(i^*(x)) - \frac{1}{2} (b_x(i) + b_x(j))$.
>
> Because the Borda score aggregates comparisons against all other arms, we prove that the resulting PM game is globally observable with parameter $k_\Pi = K-1$.
>
> The weight function $w_e$ for any two neighboring Pareto optimal actions $a = \{i^\ast, p\}$ and $b = \{i^\ast, q\}$ is defined as:
>
> $w_e(c, \sigma) = -\frac{1}{2} \left[ w^{(p)}(c,\sigma) - w^{(q)}(c,\sigma) \right]$.
> Here, $w^{(i)}(c, \sigma) = \frac{1}{K-1}$ if $c = \{i, \ell\}$ for some $\ell$ and $\sigma$ indicates $i$ wins, and $0$ otherwise.
>
> This construction satisfies the global observability condition in Definition 1 of Tsuchiya et al (2023).
>
> Specifically, $\sum_{c=1}^{k} w_e(c, \Phi_{c,x}) = L_{a,x} - L_{b,x}$ for all $x \in [d]$, as required by equation (1) in their paper.
>
> And $c_G = \max\{1, k \|G\|_\infty\} \leq \binom{K}{2} \cdot 1/(2(K-1)) \approx K/4$.
>
> Applying the global-observability guarantees of Tsuchiya et al (2023) to our reduced game yields regret bounds of $O(K^{4/3}T^{2/3}\log T)$ for the adversarial setting and $O(K^2\log^2 T/\Delta_{\min}^2)$ for the stochastic setting. Both bounds incur an additional multiplicative factor of $K$ compared with our tailored results for DB with a Borda winner. Thus, while partial monitoring offers a useful conceptual perspective, applying its general guarantees to the Borda dueling bandits problem results in strictly suboptimal results.
>
>
> 2. Comparison of algorithms and analyses.
>
> Our approach also differs fundamentally from that of Tsuchiya et al (2023). Although both works employ FTRL-style algorithms, the choice of regularizer and the resulting analysis are substantially different. In globally observable PM games, the feedback matrix $\Phi$ provides strong stability guarantees: loss differences can be estimated with low variance, quantified by the game-dependent constant $c_G$. This structure allows Tsuchiya et al to use the standard Shannon entropy regularizer to control the key intermediate term $\langle \hat{y_t}, q_t - q_{t+1} \rangle - D_t(q_{t+1}, q_t)$ as demonstrated in the proof of their Lemma 11. In contrast, the Borda dueling bandits problem does not possess such global stability properties. The Borda score depends on comparisons with all other arms, and there is no analogue of the PM constant $c_G$ that would enable the same Shannon-entropy–based argument. As a result, the analytical steps used in Tsuchiya et al cannot be directly replicated in our setting. To overcome this problem, we employ the hybrid entropy regularizer to derive the intermediate upper bound shown in the proof of Lemma 3 of our paper. This lemma is essential for both our adversarial and stochastic analyses and is not available under the PM framework.
>
> Due to space constraints, we have added the above discussion to the Appendix in the revised version. We will also incorporate it into an appropriate place in the main text in a subsequent revision.
> This reply continues in the next comment due to length restrictions.

---

> > ### Author Response · Authors · 2025-11-20
> >
> > (Continued from previous comment: Response to Reviewer zTzf) We now proceed to compare our work with Kong et al. and Ito et al.)
> > - Comparison with Kong et al., ICML 2022 and Ito et al., NeurIPS 2022
> >
> > We study a different setting from Kong et al. (2022), and our method is also of a different nature. Kong et al. (2022) investigate online learning with general graph feedback under a best-of-both-worlds framework, adopting a detect-then-switch paradigm: the algorithm initially runs in a stochastic mode and switches to an adversarial algorithm once a predefined detection condition is triggered. In contrast, our approach does not partition the environment into phases. Instead, we employ a single unified FTRL-based algorithm that automatically adapts to different regimes. Moreover, while their method provides guarantees in purely stochastic and purely adversarial environments, our algorithm additionally achieves regret guarantees in the intermediate corrupted-stochastic regime.
> >
> > Ito et al. (2022) studied the graph feedback problem using the FTRL framework. In their problem, each round reveals multiple arms' absolute losses via fixed graph neighborhoods. Their learning-rate scheduling, multi-set graph partitions, and composite regularizers are carefully designed to exploit this multi-observation structure. Crucially, their recursive regret decomposition relies on bounded estimators whose variance scales with neighborhood size, a condition guaranteed by their graph-based observability. However, the dueling bandits setting with a Borda winner provides only a single pairwise Bernoulli feedback. As a result, directly applying their single-arm or neighborhood-based estimators leads to unbounded variance. This violates the boundedness conditions required for self-bounding analysis (specifically, $|s_t(i)| \lesssim 1/K$ and $\hat{u}_t(i) \lesssim K/\delta_t^2$), preventing the recursive regret decomposition from converging.
> >
> > Our method adopts the Borda winner estimator to “symmetrically dilute” the single Bernoulli feedback. It ensures the required boundedness conditions. These guarantees allow the self-bounding inequality and mathematical inequality $\zeta(x) \leq x^2/2$ to successfully control the stability term and complete the regret decomposition in Lemma 4. Moreover, the Tsallis entropy in Ito et al.’s composite regularizer, introduced specifically to counteract graph-induced variance, has no counterpart under Borda feedback. The dueling setting does not exhibit the hybrid-observability structure that motivated this regularizer, making the entire Tsallis term theoretically irrelevant.

---

### Official Review · Reviewer_57Uf · 2025-11-01

**Soundness:** 3
**Presentation:** 3
**Contribution:** 3
**Rating:** 6
**Confidence:** 3

**Summary:**

This paper considers a relaxed comparator for setting up the regret where a uniformly winning arm is not achievable.

**Strengths:**

- The relaxed regret benchmark makes sense and the regret bounds derived therein is meaningful.
- The best-of-both-worlds results are of relevance when the underlying environment is uncertain.

**Weaknesses:**

- While it makes sense to assume the environment is static and is either stochastic, corrupted, or adversarial, I wonder if it makes more sense to assume that the environment may switch among these three regimes as the number of rounds increases. This is for example considered in Cortes, C., DeSalvo, G., Gentile, C., Mohri, M., & Yang, S. (2018, July). Online learning with abstention. In International conference on machine learning (pp. 1059-1067). PMLR.
- One of the motivation to consider dueling bandits is RLHF. I wonder how would the borda winner setting be better elaborated in such settings? And how are computational efficiency be guaranteed beyond being a convex optimization problem, that is, is there any scalable method to use for solving FTRL problems in huge scale?

**Questions:**

See the previous section

---

> ### Author Response · Authors · 2025-11-20
>
> We thank the reviewer 57Uf for the valuable comments and suggestions. Please find our response below.
>
> - Changing environment regimes
>
> Thank you for this insightful comment. We carefully revisited Cortes et al. (2018), but we did not find evidence that their framework models switching among different environment mechanisms (stochastic, corrupted, adversarial). The “non-stationarity’’ considered in their work arises from a time-varying feedback graph, while the reward-generation mechanisms are analyzed under two fixed and known settings: adversarial and stochastic.
>
> Several existing models do capture certain types of non-stationarity, but each within a single regime. For example, non-stationary bandit models allow stochastic reward distributions to drift over time, corrupted bandit models assume stochastically generated rewards with occasional perturbations, and adversarial bandit models permit arbitrary reward sequences. These frameworks treat the entire interaction as belonging to one regime throughout the horizon. To the best of our knowledge, no work considers switching among stochastic, corrupted, and adversarial regimes and provides segment-wise regret guarantees.
>
> Extending these single-regime frameworks to a setting where the environment may switch between fundamentally different mechanisms introduces substantial modeling and technical difficulties. A key challenge is how to meaningfully decompose the horizon into regime-specific segments. When observed rewards change, it is inherently ambiguous whether this reflects (i) a stochastic–stochastic switch with different means, (ii) the onset of a corrupted segment, or (iii) behavior that should be regarded as adversarial. These interpretations correspond to very different regret benchmarks. Algorithmically, this ambiguity makes it challenging to determine both when a regime switch has occurred and which regime the environment has transitioned into. Designing algorithms that can reliably identify regime boundaries and adjust their strategies accordingly, while still achieving near-optimal regret within each segment, requires techniques that go well beyond those developed for standard non-stationary or single-regime settings.
>
> Our present work focuses on achieving near-optimal performance under an unknown but fixed environment regime, which is the conventional objective in this literature. Extending the analysis to fully dynamic settings with regime switching and obtaining near-optimal guarantees for each segment is beyond the scope of this paper. Nevertheless, we agree that this is an interesting research direction and will consider it as future work.
>
>
> - RLHF settings
>
> In the tabular MDP setting, Tsuchiya et al. (2025, Reinforcement Learning from Adversarial Preferences in Tabular MDPs) provide an explicit treatment of Borda winners in an RLHF context. In their framework, the policy selects two actions at each state and receives relative feedback indicating which action wins. The Borda winner at a state is then defined as the action with the highest average winning probability against all other actions. Their formulation reduces to exactly the same structure as ours when the tabular MDP is restricted to a single-state, dueling-bandit instance.
>
> For practical RLHF settings, the learner typically receives relative feedback between two full trajectories, and preferences are not modeled directly at the level of policy pairs. Instead, it is assumed that pairwise comparisons are generated from an underlying latent-score or latent-reward model. Under commonly used preference models such as the Bradley–Terry (BT) and Plackett–Luce (PL) models, the probability of preferring one action (or policy) over another is a monotone function of their latent value differences. Consequently, the policy that maximizes the latent score automatically coincides with the Borda winner, and no additional elaboration on the Borda notion is required in these settings.
>
> Due to the character limit per reply, we have split our response. Please see our next comment for the continuation regarding computational scalability.

---

> ### Author Response · Authors · 2025-11-20
>
> (Continuation of response to Reviewer 57Uf) Below, we address the question on computational scalability and future work in this direction.
> - Scalable method
>
> When no structural assumptions are imposed on the reward function or on the action set, as in our considered tabular setting with a finite set of $K$ arms, each FTRL update amounts to solving a convex optimization problem over the $K$-dimensional probability simplex. In this case, the per-round computational cost is $\mathrm{poly}(K)$, which is standard and essentially unavoidable in the absence of additional structure on the action space.
>
> In many large-scale applications, it is common to introduce structure to reduce both computational and statistical complexity. A common approach is to represent each action $a$ with a feature vector $\phi(a)\in\mathbb{R}^d$ and assume a linear (or more general parametric) reward model in these features. In such settings, FTRL-type algorithms are implemented in the feature space: the optimization is carried out over a $d$-dimensional parameter vector. This yields algorithms whose per-round complexity scales polynomially with $d$, rather than with the number of actions $K$, and thus remains tractable even when the action set is extremely large or infinite (e.g., Abernethy et al., COLT 2009, Competing in the Dark: An Efficient Algorithm for Bandit Linear Optimization). We leave scalable BoTW dueling-bandit methods for large action spaces to future work.

---

### Official Review · Reviewer_bvPj · 2025-11-07

**Soundness:** 3
**Presentation:** 3
**Contribution:** 3
**Rating:** 6
**Confidence:** 3

**Summary:**

This paper studies the "best-of-three-worlds" problem in dueling bandits, where the goal is to attain regret bounds in adversarial, stochastic and corrupted problem simultaneously. In particular, they show such guarantees in terms of the Borda regret, which is the cumulative difference in Borda score between the played arms and hte Borda-optimal pair of arms. The proposed algorithm for this task estimates the pairwise probabilities using so-called importance weighted estimators, and then runs a regularized-follow-the-leader (RFTL) algorithm using these estimators. The output of RFTL is then mixed with a uniform distribution to get the policy for a given round.

**Strengths:**

- To my knowledge, the best-of-three-worlds with Borda score is a new contribution.
- The regret bounds for the adversarial and stochastic settings match the optimal regret in those settings up to log factors. (Per the lower bounds in Saha et al., 2021).
- The paper gave a clear explanation of how the analysis of best-of-three worlds with Cordocet regret does not transfer to Borda regret, and therefore they require a new algorithm/analysis approach.

**Weaknesses:**

- Although the best-of-three worlds guarantees are new for this setting, the actual individual guarantees do not necessarily improve on those that have been attained previously. In particular, the guarantees on the corrupted setting are $1/\Delta^2$, while prior work (Agarwal et al, 2021) showed $1/\Delta$ dependence. This renders the claim on line 99 incorrect, that the guarantees match the state-of-the-art for the corresponding setting.
- I found the presentation to somewhat confusing at places, which I have detailed in the Questions box.

**Questions:**

- Should the $R_T$ in (1) be $R_T^b$? The environment definitions that follow use $w(i)$, which makes it seem that it is in reference to the shifted regret $R_T^b$.
- In line 203, it appears that the expectation is over the $w$, but the expectation is written as $E_{s \sim D}$. I found this to be confusing, and would suggest that the authors modify for clarity.
- Equation (2) uses $u_K$, but it hasn't been defined yet.
- In line 242, it says that $x_t$ and $y_t$ are sampled from $d_t$, but the Algorithm 1 shows them being sampled from $\pi_t$.
- I would suggest adding an intuitive explanation for the algorithm step sizes in (5).

---

> ### Author Response · Authors · 2025-11-20
>
> We thank the reviewer bvPj for the valuable comments and suggestions. Please find our response
> below.
>
>
> - Comparison with Agarwal et al. (2021) in Table 1
>
> We thank the reviewer for pointing this out. Indeed, that work studies dueling bandits under the Condorcet-winner objective and is therefore not directly comparable to our results for the more challenging Borda-winner setting. We have removed this line from the table in the revised version.
>
>
> - Typos
>
> We thank the reviewer for carefully identifying these typos.
> We have corrected them in the revised manuscript.
> Specifically, $R_T$ in (1) has been updated to $R_T^b$.
> In line 203, the expectation is now written as $\mathbb{E}_{w \sim D}$.
> In equation (2), we have added an explanation that $u_K$ denotes the uniform distribution with $u_K(i) = 1/K$ for all arms $i$.
> In line 242, $d_t$ has been corrected to $\pi_t$.
>
>
>
> - Intuition on the step size
>
> Overall, the choice of step sizes must allow the algorithm to adapt automatically to the underlying environment. By optimizing the upper bound in Lemma~4, our goal is to achieve a $T^{2/3}$ regret upper bound in the adversarial setting and a $\log T / \Delta^{2}$ upper bound in the stochastic setting. To enable such environment-dependent behavior, our step sizes are designed to depend on $v_t$ and $e_t$.
>
> In the adversarial case, each arm always retains a non-zero probability of being selected, and thus $v_t$ and $e_t$ can be regarded as constants. According to equation(11) in Lemma4, we need to set $\delta_t \sim  t^{-1/3}$, $\alpha_t \sim t^{2/3}$, and $\alpha_{t+1} - \alpha_t \sim t^{-1/3}$ in order to obtain the overall $T^{2/3}$ regret bound. In the stochastic case, the probability distribution over the arms gradually shifts from being diffuse to being concentrated on the optimal arm; consequently, $v_t$ and $e_t$
> decrease from non-zero values and eventually converge to zero. To derive the $\log T / \Delta^{2}$ bound, we select $\alpha_t$ and $\delta_t$ so that the regret can be controlled
> through the $\bar{S}^{2/3}$ term. Using the inequality $x^{1/3} y^{2/3} \le \tfrac{1}{3} x + \tfrac{2}{3} y$ in equation~(21) yields the desired stochastic bound of $\log T / \Delta^2$.
>
> We have also incorporated this discussion into the revised manuscript.

---

### Author Response · Authors · 2025-12-02
**Summary of Author Responses**

Dear AC,

We sincerely thank all reviewers for their insightful and constructive feedback, which has significantly strengthened the theoretical rigor and contextual grounding of our paper. All three reviewers provided positive evaluations, highlighting the submission’s soundness, presentation quality, and contribution. The primary suggestions centered on expanding the related-work discussion and elaborating on potential extensions.

In response, we have prepared detailed point-by-point rebuttals and revised the manuscript accordingly. We believe these changes substantially improve the paper’s positioning, clarity, and technical depth.

Summary of Author Responses

1. Related work and technical differences (Reviewer zTzf)

   We have expanded our discussion of recent best-of-both/three-worlds algorithms in partial-monitoring and graph-feedback settings. In particular, we now (i) provide an explicit reduction from Borda dueling bandits to a globally observable partial-monitoring game; (ii) show that generic PM guarantees are suboptimal by a factor of (K); and (iii) explain why our hybrid-entropy FTRL method with Borda-specific estimators is necessary. We also clarify the technical distinctions between our work and these prior approaches.

2. Algorithmic intuition and design choices (Reviewer bvPj)

   We have added an intuitive explanation of the adaptive step sizes and hybrid regularizer, illustrating how they automatically yield $T^{2/3}$ regret in adversarial regimes and $O\left(\frac{\log T}{\Delta^2}\right)$ regret in stochastic regimes. This discussion is now included in the main paper to improve accessibility.

3. Broader applicability and future directions (Reviewer 57Uf)

   We elaborate on the relevance of Borda winners in RLHF settings and discuss the modeling and algorithmic challenges in regime-switching environments proposed by the reviewer.

4. Presentation and minor issues (Reviewers bvPj)

   All identified typos, notational inconsistencies have been corrected throughout the manuscript.

We again thank the reviewers for their constructive and positive evaluations, and we welcome further discussion.

Best regards,
The Authors of Submission 18375

---

### Public Comment · ~Zirui_Hu3 · 2026-04-15
**Official Withdrawal Statement: This Paper Has Been Withdrawn by Authors Due to Critical Bug**

We officially confirm that this submission has been withdrawn.

After careful examination, we discovered a critical bug in the paper that is not easy to correct. Therefore, we have made the cautious decision to withdraw the submission.

We sincerely apologize for any inconvenience this may cause to the reviewers, area chairs, and program chairs who have invested their time in evaluating our work. We are deeply grateful for all the valuable feedback and efforts from the ICLR community.

Best regards,
Zirui Hu and Fang Kong

---

### Note · Authors · 2026-03-02

I have read and agree with the venue's withdrawal policy on behalf of myself and my co-authors.

---

> ### Note · Program_Chairs · 2026-03-31
>
> We approve the reversion of withdrawn submission.

---

### Note · Authors · 2026-06-05

I have read and agree with the venue's withdrawal policy on behalf of myself and my co-authors.

---

### Meta-Review · Area_Chair_oK2S · 2026-01-02

**Summary:**

All reviewers viewed the paper positively and rated it marginally above the acceptance threshold. They agreed that the paper makes a novel contribution by establishing best-of-three-worlds guarantees for dueling bandits under the Borda winner objective, which requires new algorithmic and analytical techniques beyond prior work on Condorcet winners. The main concerns were about clarity, positioning with respect to related work, and the accuracy of some claims, rather than about the correctness of the results.

**Reviewer Concerns:**

The rebuttal addressed the reviewers’ main concerns. Issues regarding an inaccurate comparison to prior work and several presentation and notation problems were corrected, and additional intuition was provided for the algorithmic design choices. Questions about the novelty of the partial monitoring and graph-feedback frameworks were answered with a detailed technical comparison, clarifying why generic approaches would yield suboptimal bounds and why the proposed hybrid regularizer is necessary. Broader questions about regime switching, RLHF motivation, and scalability were addressed through expanded discussion and are reasonably treated as scope or future-work considerations. No major unresolved concerns remain.

**Reviewer Scores:**

One reviewer would likely increase their score slightly given that their main substantive and presentation-related concerns were fully resolved. The other two reviewers would likely keep their scores unchanged or possibly increase them slightly, as their questions were primarily about scope, positioning, and clarification rather than fundamental issues with the results.

---

### Decision · Program_Chairs · 2026-01-26

Accept (Poster)